# MammoGAN:
# High-Resolution Synthesis of Realistic Mammograms

**Dimitrios Korkinof** [1]                                    DIMITRIOS@KHEIRONMED.COM

**Andreas Heindl** [1]                                        ANDREAS@KHEIRONMED.COM

**Tobias Rijken** [1]                                          TOBIAS@KHEIRONMED.COM

**Hugh Harvey** [1]                                           HUGH@KHEIRONMED.COM

**Ben Glocker** [1,2]                                    B.GLOCKER@IMPERIAL.AC.UK

[1] *Kheiron Medical Technologies Ltd., London, UK*

[2] *Department of Computing, Imperial College London, London, UK*

**Editors:** Under Review for MIDL 2019

## Abstract

We explore whether recent advances in generative adversarial networks (GANs) enable synthesis of realistic medical images that are hard to distinguish from real ones, even by domain experts. High-quality synthetic images can be useful for data augmentation, domain transfer, and out-of-distribution detection. However, generating realistic images is challenging, particularly for Full Field Digital Mammograms (FFDM), due to the high-resolution, textural heterogeneity, fine structural details and specific tissue properties. We employ progressive GANs to synthesize mammograms at a resolution of 1280x1024 pixels, the highest reported so far. In order to assess the perceptual realism, we designed a user study where experts are asked to distinguish real and generated images with exciting results.

**Keywords:** Image Synthesis, Generative Adversarial Networks, Breast Mammography

## 1. Introduction

The generation of synthetic medical images is of increasing interest to both the medical and machine learning communities for several reasons. First, synthetic images can be used to improve methods for downstream detection and classification tasks, by means of data augmentation (Salehinejad et al., 2018; Frid-Adar et al., 2018). Second, image-to-image translation can be used for domain adaptation (Kamnitsas et al., 2017), image enhancement (Yi and Babyn, 2018) and super-resolution (Ledig et al., 2017).

For Full Field Digital Mammograms (FFDM), it is important to view the images in high-resolution due to fine structural details of high diagnostic importance, such as lesion spiculation and micro-calcifications. However, generating synthetic medical images in high-resolution is particularly difficult due to the dimensionality of the output pixel space.

In this work, we explore the use of progressively trained GANs (Karras et al., 2018) and show that generation of realistic, high-quality synthetic mammograms is feasible at a resolution up to 1280x1024 pixels. To the best of our knowledge, this is the highest resolution achieved for medical image synthesis to date. Our user study where we ask experts to distinguish between real and generated images further confirms the high quality synthesis. We hope this work can inspire further research on GANs in the medical imaging domain.

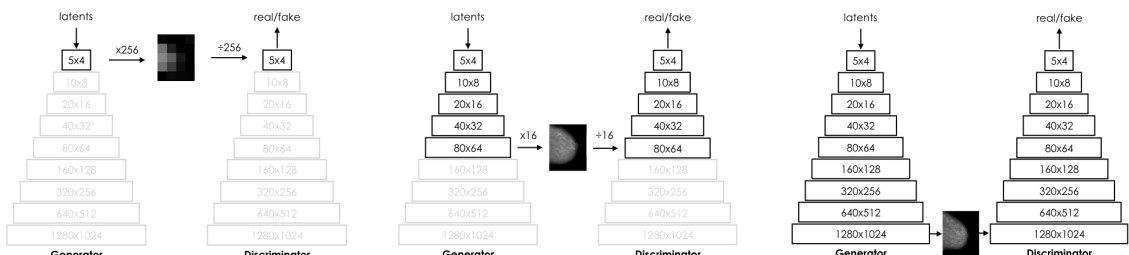

Figure 1: Illustration of progressive training at 3 different scales.

## 2. Data and clinical setting

Mammograms are soft tissue breast X-rays acquired after each breast, in turn, has been flattened using two plastic paddles. Conventionally, both left and right breasts are imaged using two standard views, the cranial-caudal (CC) and the mediolateral-oblique (MLO).

Our training dataset consists of more than 1 million mammograms, from which we have excluded images containing post-operative artifacts (metal clips, etc.) as well as large foreign bodies (pacemakers, implants, etc.). After applying aspect ratio preserving image resizing, the final resolution is 1280x1024 pixels.

## 3. Method

A key development for scaling GANs to higher resolutions is progressive training, as originally proposed in (Karras et al., 2018). The main concept is to start training at a very low resolution, before gradually increasing it as more layers are phased in (Fig. 1). For further stability and performance gains, it is common practice to use this strategy in combination with a Wasserstein objective and gradient penalty, as suggested in (Gulrajani et al., 2017).

Despite using progressive training, we still experienced stability issues, which we alleviate by: a) Adding supervised information, as suggested in (Salimans et al., 2016). We conditioned on useful attributes, such as the view (CC or MLO), breast density and breast mask size; b) Decreasing the default learning rate (from 0.002 to 0.0015); c) Gradually increasing the discriminator iterations per each generator update, from 1 to a maximum of 5; d) Increasing the capacity of the networks, by doubling the number of neurons in the final feature layer and the starting depth of the network. We trained for 33 epochs, before resuming and presenting an additional 5 million images. We selected the best network checkpoint based on the sliced Wasserstein distance (Karras et al., 2018). The whole process took approximately 52 hours on an NVIDIA DGX-1, with 8 V100 GPUs.

## 4. Synthesis Results

Samples drawn from our trained MammoGAN are generally of high quality and variability. The MLO view appears more difficult showing more artefacts. Common failures include dark or bright spots, stretching and failed reproductions of breast implants, a few of which have been accidentally included in the training. Based on a sample of 1000 randomly generated synthetic MLO views, we observed that 86.4% of the images had no visible artefacts. Examples are shown in Fig. 3 and in our technical report (Korkinof et al., 2018).

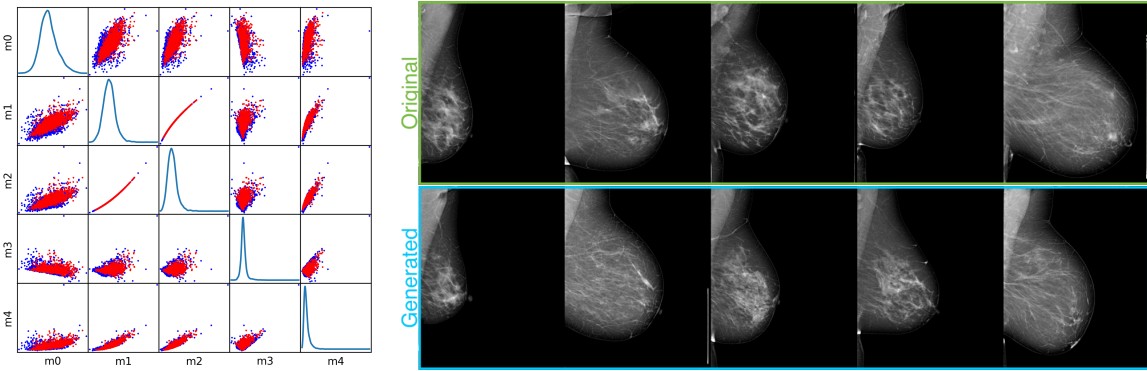

Figure 2: First five statistical moments, red denoting real and blue synthetic images.

Figure 3: Randomly sampled examples of real and generated MLO views.

An interesting finding is that the generator resists to reproduce structures such as calcifications and external skin markers, which both appear as bright spots. We hypothesize that the network architecture acts as a prior preventing these features from being captured, as suggested in (Ulyanov et al., 2018). More results are in (Korkinof et al., 2018).

## 5. Quality Assessment and Conclusion

To assess the quality of the generated images we rely on two methods: First, we plot the first five statistical moments (mean, variance, skewness, kurtosis and hyperskewness) to assess similarity between low-level pixel distributions of real and synthetic images (see Fig. 2). We observe that the image statistics overall correspond reasonable well while this mechanism may allow to prune outliers from the synthesis which will be explored in future work. Other evaluation metrics (i.e. Frechet inception, sliced Wasserstein etc.) may be considered but are mostly useful to compare synthesis methods with each other but not to evaluate the human perception and perceived realism by domain experts. To this end, we conducted a randomised user study to determine whether synthetic images can be distinguished from real ones as a proxy for perceptual realism. Specifically, we randomly selected 1000 synthetic and 1000 real MLO views and excluded images with visible artefacts (13.6% from the synthetic; 2.8% from the real). Randomly assigned real/synthetic image pairs were displayed in a custom tablet application built in Unity, with image pinch and zoom capability. A total of 117 attendees at the Radiological Society of North America conference were each asked to assess 10 randomly-paired images with no time limit. User had to select which of the two presented images was real. 55 were radiologists (82% board certified, 60% specialised in breast radiology) and 62 were not. A binomial distribution with success probability $\pi = 0.5$ ($p = 0.999$, Chi-square test) was observed indicating that the participants (including domain experts) were unable to distinguish between generated and real mammograms confirming high quality synthesis. From this study we conclude that recent advances in GAN-based image synthesis can be successfully translated to high-resolution medical images. Open challenges remain in reproducing specific pathological patterns of diseases and the demonstration of improved downstream analysis when using synthetic images for data augmentation.

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
