# OpenReview forum: "MammoGAN: High-Resolution Synthesis of Realistic Mammograms"
_MIDL.io/2019/Conference/Abstract — MIDL Abstract 2019_

### Official Review · AnonReviewer2 · 2019-04-28
**Evaluation of progressive GANs for mammography synthesis**

**Rating:** 3
**Confidence:** 2

**Review:**

Authors demonstrate how to generate realistic mammography images at a high resolution of 1280x1024 pixels, the highest so far reported in medical imaging.

Realistic and novel evaluation framework ran during RSNA meeting particapnts.

Results are well analyzed and discussed, which is expected to be of interest to attendees of MIDL

---

### Official Review · AnonReviewer1 · 2019-05-01
**Synthesizing mammograms at a high resolution with a progressive growing of GANs**

**Rating:** 3
**Confidence:** 2

**Review:**

The authors propose to use a progressive growing of GANs (Karras et al. ICLR'18) to synthesize mammograms at a high resolution. They further report a user study, in which experts were asked to distinguish real and generated images. The results are interesting.

This is a well-written and clear paper. The technical novelty limited (ideas of Karras et al., ICLR'18), but the application and obtained results are interesting. Synthesizing images at a resolution of 1280 x1024 pixels is the highest reported so far in the context of GANs (the authors made several ad hoc choices for this to work nicely).

---

### Decision · Program_Chairs · 2019-05-06
**Acceptance Decision**

Accept